# OpenReview forum: "Unifying Adversarial Robustness and Training Across Text Scoring Models"
_ICML.cc/2026/Conference — ICML 2026 regular_

### Official Review · Reviewer_GJhu · 2026-02-17

**Soundness:** 1
**Presentation:** 1
**Significance:** 1
**Originality:** 3
**Overall Recommendation:** 1
**Confidence:** 5

**Summary:**

This work studies the adversarial robustness of text scoring models. Authors denominate retrievers, rerankers and reward models as text scoring models. Authors argue that, while text scoring models are all based on the same foundations, their robustness is studied focused on the task or on a specific attack. Additionally, authors argue that, unlike open-ended text generation, where defining the success of an attack is difficult, scoring models provides a neat definition of an attack success. Authors perform adversarial training of text scoring models with Rudimentary, PGD, HotFlip attacks or a combination of them. Authors find that robustness does not transfer between threat models and find that adversarially trained reward models provide a higher performance when used for RLHF training of LLMs and testing on Arena-Hard and WildBench.

**Compliance With Llm Reviewing Policy:**

Affirmed.

**Key Questions For Authors:**

- Which model do you use for the judge in Table 3?

- Your adversarial training method seems rather fast (your longest run is 36h). Do you attack every sample and then train, or do you attack on the fly every batch? I presume PGD will be the faster to train, how does it compare against the Rudimentary and HotFlip perturbations?

- Are classifiers text scoring methods?

**Limitations:**

Authors do not describe the limitations of their method. At the moment they are difficult to understand with no pseudocode and no equations defining the problem they solve.

**Strengths And Weaknesses:**

**Strengths**

- Authors evaluate multiple threat models and test how robustness transfers when training/testing with different threat models.

- Authors show that employing adversarially robust reward models improves the performance of models trained through RLHF with those robust models.


**Weaknesses**

While the insight about using robust reward models is quite interesting, I believe many things could be improved about the paper.

- **Unclear motivation**
	1) Authors argue that robustness is studied focusing too much on specific attacks or tasks and that a unifying view is needed. While I agree that we would like to understand how to make LLMs (or any text model) robust to perturbations of any (reasonable) kind, adversarial attacks are inevitably designed for specific tasks. While GCG and HotFlip are both gradient based attacks, their purpose is completely different. GCG appends a sequence of tokens to some prompt in order to elicit Jailbreaking. Hotflip is thought to apply semantics-preserving perturbations to a sentence in order to produce missclasification. Most importantly, GCG can introduce ANY token in the suffix, HotFlip only replaces synonyms in the given sentence. While you could apply GCG for classification or HotFlip for Jailbreaking, their success would be limited and more importantly, GCG would not preserve the meaning of the input sentence.
	2) Authors argue that defining what is a successful attack is much easier for scoring models. First, wasn't this solved already by doing adversarial attacks on classification models? Second, according to your definition, whouldn't classification models be text-scoring models?

- **Missing a lot of key details about the method and experimental setup** The setup in this work is non-standard: 1) Authors introduce a "Rudimentary" attack that is briefly mentioned to apply insertions, deletions and swaps (replacements?) of characters or words. Authors also mention that all of the rudimentary, HotFlip or MLM perturbations are optimized through beam search in Section 4.1. However, little details about this beam search approach are provided, a pseudocude and a comparison against standard HotFlip or other attacks would help understand the benefits of using this attack.
2) Authors introduce a new adversarial training variant with a combination of multiple threat models and some key loss terms as described in lines 220-229. However, no pseudocode of the overall method or equations describing their main optimization problem are described.

- **Missing a lot of references** Multiple related works are missing citations/comparison. Regarding the adversarial attack used for training, [1] previously studied beam search attacks for Jailbreaking LLMs. Regarding the rudimentary perturbations, looks like the perturbations you apply are constrained based on the Levenshtein distance [2] on tokens or directly on characters. On this regard, there are multiple works applying threat models like yours: DeepWordBug [3], TextBugger [4], Pruthi et al., [5], Charmer [6] or CWBA [7]. While it is not super important, your PGD training strategy: single step and random initialization, is known as RS-FGSM [8]. While you already discuss some adversarial training defenses, I missed ASCC [9] and the ones in [10,11,12]. In the introduction, you jump from the adversarial perturbations on images from (Szegedy et al., 2014) to the jailbreaking attacks of (Zou et al., 2023), I think it is worth mentioning early works on adversarial perturbations in the text domain: [13,14]. In lines 133-135 you mention "we are unaware of work explicitly adversarially training retrievers or rerankers for robustness to GCG/HotFlip-style attacks." while I am not aware either of methods doing it for GCG or Hotflip, there are works training robust text encoders (could be used for retrieval or reranking) against character-level perturbations [15]. As stated in lines 83-85, one of your contributions is that "This work provides the first evaluation of adversarial training robustness transfer across attacks on these scoring models.", while your definition of "text scoring" is not clear, if we assume classifiers are text scorers, [16] (which you cite), [6] and [12] already study this and arrive to similar conclusions as you.

In general, if I were to rewrite the paper, I would focus on the RLHF experiments. I feel there is real value there and providing more insights on the effect of robust reward models on the downstream performance would be super interesting.


**References**

[1] Sadasivan et al., Fast Adversarial Attacks on Language Models In One GPU Minute. ICML 2024

[2] Levenshtein, Binary codes capable of correcting deletions, insertions, and
reversals. Soviet Physics Doklady 1966.

[3] Gao et a., Black-box generation of adversarial text sequences to evade deep learning classifiers. IEEE SPW 2018.

[4] Li et al., Textbugger: Generating adversarial text against real-world applications. NDSS 2019.

[5] Pruthi et al., Combating adversarial misspellings with robust word recognition. Annual Meeting of the ACL 2019.

[6] Abad Rocamora et al., Revisiting Character-level Adversarial Attacks for Language Models. ICML 2024.

[7] Liu et al., Character-level white-box adversarial attacks against transformers via
attachable subwords substitution. EMNLP 2022.

[8] Wong et al., Fast is better than free: Revisiting adversarial training. ICLR 2020.

[9] Dong et al., Towards Robustness Against Natural Language Word Substitutions. ICLR 2021

[10] Miyato et al., Adversarial Training Methods for Semi-Supervised Text Classification. ICLR 2017.

[11] Wang et al., InfoBERT: Improving Robustness of Language Models from An Information Theoretic Perspective. ICLR 2021.

[12] Hou et al., TextGrad: Advancing Robustness Evaluation in NLP by Gradient-Driven Optimization. ICLR 2023.

[13] Belinkov and Bisk, Synthetic and natural noise both break neural
machine translation. ICLR 2018.

[14] Alzantot et al., Generating natural language adversarial examples. EMNLP 2018.

[15] Abad Rocamora et al., Robustness in Both Domains: CLIP Needs a Robust Text Encoder. NeurIPS 2025.

[16] Morris et al., TextAttack: A Framework for Adversarial Attacks, Data Augmentation, and Adversarial Training in NLP. EMNLP 2020.

---

> ### Author Rebuttal · Authors · 2026-03-29
>
> We address key points below.
>
> **GCG vs. HotFlip and motivation.** GCG and HotFlip both apply gradient-guided discrete token-swap attacks. To be clear, HotFlip is not synonym-only or semantics-preserving. Similar to GCG, it can introduce any token through a swap. In the original paper, the core operation is a gradient-based swap; semantic constraints are an optional add-on, not inherent to the method, and often do not hold in practice. In our setting, and in prior ranking work such as Zhong et al. (2023), HotFlip-style edits are often clearly not semantics-preserving, as our appendix figures (Figures 3–6) show. The point of our unified view is not to collapse all attacks into one, but to study whether robustness transfers across related attack families and across LM roles that all score text. These attacks all share the same high-level goal: making the model score irrelevant or rejected text above relevant or chosen text.
>
> **Attack success and classifiers.** We focus on broadly capable LM-based scorers used in modern pipelines: dense retrievers, rerankers, and reward models. Retrievers and rerankers operate over large corpora and diverse queries, while reward models evaluate responses across diverse prompts and criteria such as relevance, clarity, style, and safety. By contrast, classifiers are much narrower task-specific models, often focused on tasks like sentiment or news classification. We do not consider classifiers text scorers within our study. We can clarify the wording in revision.
>
> **RLHF focus.** We appreciate the suggestion to focus on the RLHF experiments. We have expanded this evaluation with broader alignment benchmarks (OR-Bench, XSTest, WildGuardMix) judged by Nemotron 3 Super (120B), and a length-controlled analysis of Arena-Hard and WildBench. Adversarially trained RM policies are consistently preferred over base RM policies on safety benchmarks, and this preference holds even when the base-RM policy produces longer responses, arguing against judge length bias. See our response to reviewer ivJn for full tables. We believe these results, combined with the KL-divergence analysis in the paper, provide strong evidence that adversarially robust reward models meaningfully improve downstream alignment.
>
> **Beam-search attack details.** Section 4.1 specifies the full procedure: 16 beams, 16 proposals per beam item, up to 512 steps, and retention of up to 8 previous beam items. We agree a compact pseudocode box could aid readability and will add one, but the method is specified and our code is provided for reproducibility.
>
> **Combined training method.** Similarly, Section 4.2 describes all loss terms and the full training setup, and Appendix D provides hyperparameters. Pseudocode would improve readability, but the method is specified and reproducible as written.
>
> **Beam search for jailbreaks.** BEAST is relevant related work and we will cite it. Our use of beam search, however, is not attack-specific novelty; it is simply the search procedure we use across perturbation attacks.
>
> **Rudimentary perturbations.** Our threat model is not a fixed Levenshtein ball. Edits are applied iteratively, can overwrite or modify previous edits, and include both character-level and word-level operations.
>
> **RS-FGSM vs. PGD.** FGSM takes a step along the sign of the gradient. Our training uses a random-start projected gradient step in continuous embedding space. PGD is the more accurate description.
>
> **Training efficiency.** We generate adversarial perturbations on the fly for each batch. Appendix J reports a longest run of 40 hours for the most demanding setting (all four methods combined). Rudimentary is faster than PGD and HotFlip because it does not require gradients to generate adversarial samples.
>
> **Missing references.** None of the cited works address adversarial training of text scoring models including retrievers, rerankers, or reward models, which is our core setting. We already cite impactful NLP adversarial work (HotFlip, TextFooler, BERT-Attack, TextAttack, FreeLB, ALUM). We agree the paper could cite more for broader context and will add some of the suggested references in revision. None overlap with the novelty of our work. For example, **(Abad Rocamora et al., NeurIPS 2025).** This work adversarially finetunes CLIP's text encoder for multimodal robustness and does not focus on text-text retrieval, cross-encoder reranking, or reward modeling. Our work addresses a different setting (text scoring models) and a different question.
>
> **"First evaluation" claim.** To be precise, our claim is scoped to text scoring models: retrievers, rerankers, and reward models. While some cited works include cross-attack evaluation in classification, none evaluate adversarial training transfer for these scoring model roles, which is what we study.
>
> **Key questions.** The judge in Table 3 is Gemini 3 Flash.
>
> **Limitations Described** The paper discusses limitations in Appendix F ("Scope, Limitations, and Future Work").

---

> > ### Author Rebuttal · Reviewer_GJhu · 2026-04-03
> >
> > Dear authors,
> >
> > Thanks for your rebuttal. Unfortunately, my concerns remain unresolved.
> >
> > It is true that you are the first to study Adversarial Training (AT) for text scoring models, as far as I know, you are also the first to use such denomination. It remains unclear **what is a text scoring model** and why classifiers are not text scoring models. While the setup you are considering has not been considered before, there needs to be a connection with the previous literature to help the reader understand what are your contributions.
> >
> > As stated in my original review:
> > 1) Your "Rudimentary" threat model is non-standard. It becomes hard to perform apples-to-apples comparisons with other attacks. If you want to focus on AT and compare against other AT methods, you need to adhere to one threat model (at least during evals).
> > 2) Your beam search approach is non-standard and little is understood about it. Let's say you focus on the same threat model as HotFlip and take a fixed trained model. What is the Attack Success Rate (ASR) of your attack compared to HotFlip? What is the effect of your hyperparameters in the ASR? Section 4.1 sure reports the hyperparameters of your attack, but there is no ablation (apart from Table 8) explaining why you chose those.
> > 3) You study "Text scoring models" (retrievers, rerankers and reward models). As far as I understand, this is the first paper to study AT in this setup. But, AT has been studied in the text domain for many other tasks. Just saying that other works have not considered your setup is unfortunately not enough. For example, you could take your training objective and use the AT method of [9,10,11,12] in classification or even (Xhonneux et al., 2024)'s method for LLMs. It is true that [15] does not evaluate in text-to-text retrieval tasks, but they train text embedding models (same as you in some cases). The point is that you should be able to adapt previous methods to your setup, or adapt your method to another setup and perform comparisons.
> >
> > At the moment, the paper needs to go through another revision. I appreciate the added RLHF experiments.
> >
> > Best regards,
> >
> > Reviewer GJhu.

---

> > > ### Author Response · Authors · 2026-04-03
> > >
> > > Thank you for your continued engagement. We appreciate the opportunity to clarify the remaining concerns.
> > >
> > > **On the definition of "text scoring model" and why classifiers are excluded.**
> > >
> > > We addressed this in our rebuttal under "Attack success and classifiers,". This might have gotten hidden, so we are happy to restate it here:
> > >
> > > We focus on broadly capable LM-based scorers in modern NLP pipelines: dense retrievers, rerankers, and reward models. Retrievers and rerankers score the relevance of arbitrary text to diverse queries over large corpora. Reward models evaluate responses across diverse prompts and criteria such as relevance, clarity, style, and safety. These models share a common structure: they ingest open-ended text pairs, produce scalar scores, and must generalize across a wide distribution of inputs.
> > > Reward models are especially broad: they can be used to score responses in essentially any generative LLM input/output setting.
> > >
> > > By contrast, classifiers are narrower, task-specific models mapping inputs to a small, fixed label set (e.g., sentiment polarity, news categories). Their input distribution and output space are fundamentally more constrained. We do not consider classifiers text scoring models within our study, and we believe this distinction is natural and useful. We will clarify the terminology in revision.
> > >
> > > **On the threat model and comparability of attacks.**
> > >
> > > Our evaluations are directly comparable in the sense that all swapping/perturbation attacks in our paper, including HotFlip, Rudimentary, and MLM-guided attacks, use the same beam-search evaluation procedure and attack budget (Section 4.1). What differs is the candidate proposal mechanism: HotFlip uses gradient-guided token swaps, Rudimentary uses simple character- and word-level edits (insertions, deletions, swaps, and duplications), and MLM-guided attacks use contextual replacements. The beam search procedure, perturbation budget, and evaluation protocol are otherwise identical across these methods. Therefore, the Attack Success Rates in our results tables (e.g., Tables 1, 2, 4-6) are directly comparable under a common search budget and evaluation protocol, even though these are different attack families.
> > >
> > > **On beam search hyperparameters.**
> > >
> > > The hyperparameters we select (number of beams and number of variants per beam) are chosen to provide a strong search budget within our computational constraints. Table 8 shows that weaker search budgets can substantially understate attack success, which is why we report results with our strongest feasible setting.
> > >
> > > **On connections to prior adversarial training methods.**
> > >
> > > We appreciate the suggestion to adapt prior AT methods such as [9,10,11,12] and Xhonneux et al. (2024) to our setup. Our work is closest in spirit to these prior adversarial-training families, and we instantiate two of the same broad approaches in our text-scoring setting. The methods in [9,10,11,12] span continuous embedding-space perturbations ([10] Miyato et al.), discrete token-level adversarial training ([9] ASCC, [12] TextGrad), and information-theoretic regularization ([11] InfoBERT). In our paper:
> > >
> > > 1. Our PGD-based adversarial training uses continuous embedding perturbations, placing it in the same broad adversarial-training family studied by [10] (Miyato et al.) for classification and by Xhonneux et al. (2024) for LLMs. This is a standard AT approach that has been applied across many settings, including with NLP models (e.g., FreeLB and ALUM, which we cite), and we apply it here to text scoring models across all model types.
> > >
> > > 2. Our HotFlip-based adversarial training likewise falls in the broader family of discrete gradient-guided adversarial training, which includes methods such as [9] (ASCC) and [12] (TextGrad), adapted here to our text scoring setup.
> > >
> > > Indeed, the fact that these same fundamental techniques keep being independently applied across different NLP subfields is part of our motivation: we provide a unified study of AT approaches specifically for text scoring models, enabling the direct cross-method comparisons that have been absent. We will make these connections to [9,10,11,12] more explicit in revision.
> > >
> > > We hope this clarifies the remaining concerns and better situates our experimental design relative to prior work.

---

### Official Review · Reviewer_FVW7 · 2026-03-12

**Soundness:** 3
**Presentation:** 3
**Significance:** 3
**Originality:** 3
**Overall Recommendation:** 4
**Confidence:** 2

**Summary:**

This paper studies adversarial robustness in text scoring models (retrievers, rerankers, and reward models), where failures are measured as ranking/scoring errors. It compares several adversarial training methods, including PGD, HotFlip-style token substitution, simple string edits, paraphrase consistency, and a new content-injection method, and analyzes robustness transfer across attack types and model roles. Results show that combining complementary methods improves robustness with little or no utility drop, and adversarially trained reward models in RLHF reduce reward hacking (lower KL drift) and produce policies preferred by an LLM judge.

**Compliance With Llm Reviewing Policy:**

Affirmed.

**Final Justification:**

I decide to stay with my init score

**Key Questions For Authors:**

plz see above

**Limitations:**

yes

**Strengths And Weaknesses:**

Strength

1. The paper gives a clear and practical framing: it treats adversarial robustness in retrievers, rerankers, and reward models under one scoring-based view, so failures are defined consistently as ranking/scoring errors rather than vague semantic mismatches.

2. A strong contribution is the content-injection training setup, which targets a threat class that token-level perturbation defenses often fail to cover. This makes the method set more complete and practically relevant.

3. The experiments are fairly comprehensive: they cover multiple model roles and multiple attack families, and also analyze transfer across attacks, which is more informative than reporting gains on a single threat type.

4. The RLHF results add real application value: adversarially trainedreward models show reduced reward-hacking signals (lower KL drift) and produce policies preferred by an LLM judge, suggesting usefulness beyond isolated robustness benchmarks.


Weaknesses
1. The current evidence is mainly on relatively small scoring backbones, so it is still unclear whether the same robustness gains hold for stronger SOTA retrievers/rerankers and larger reward models.

2. Some threat-model choices are only partially realistic in deployment, especially for retrieval: query/content injection is tested in a controlled setup, but end-to-end corpus poisoning and reindexing effects are not fully covered.

3. Several evaluation settings may weaken external validity. For example, reranker robustness is measured on a fixed top-100 candidate pool from a base retriever, and reward-model tests use a small pre-filtered subset, which can bias conclusions.

4. The baseline space is not fully saturated. Comparisons to stronger robustness-oriented methods(e.g., multi-step/efficient adversarial training variants and regularization-based defenses) are limited, making it harder to position the proposed method against the best available alternatives.

---

> ### Author Rebuttal · Authors · 2026-03-29
>
> Thank you for your positive review. We address each point below.
>
> **Model scale.** Our experiments span a wide range of models: E5 BERT-base (encoder-only, ~110M params) for retrieval, Qwen3-0.6B (decoder-only) for reranking, and Llama-3.2-3B / Llama-3.1-8B (decoder-only) for reward modeling and RLHF. We also evaluate state-of-the-art Skywork-Reward-V2 models at 1B, 3B, and 8B scales in the appendix. These models differ in architecture family, attention pattern, parameter count (110M to 8B), and training recipe. Notably, hyperparameters tuned on the 3B RM transfer directly to the 8B RM with strong results. The Skywork analysis further shows that robustness does not automatically improve with scale. In fact, larger models can be more susceptible to injection attacks, reinforcing that adversarial training is needed regardless of model size. Nonetheless, evaluating larger models (30B+) is an important future direction. However, we believe we've covered a meaningfully diverse set given practical compute constraints.
>
> **Controlled threat-model setup.** This design is deliberate: it isolates model-level robustness from system-level confounds and allows consistent comparison of attacks and defenses across retrievers, rerankers, and reward models under the same conditions. The goal of model-level robustness training should be broad and effective robustness so that wherever the model is deployed, it is resilient to adversarial manipulations. Real-world attackers can try many strategies; a model that is robust in controlled evaluation is better positioned for deployment than one that has not been stress-tested or only stress-tested in constrained conditions. Our controlled setup is complementary to end-to-end evaluations, it isolates model-level robustness, which is important for system-level robustness. In a corpus poisoning scenario, the fundamental question is still whether the model scores adversarial documents above relevant ones. Our setup directly measures this.
>
> **Evaluation settings.** For rerankers, using the top-100 retrieved candidates reflects a standard reranking pipeline. Full-corpus reranking over millions of documents is computationally prohibitive and we believe re-ranking the top-100 candidates retrieved from a strong retriever is a fair setting. For reward models, the pre-filtering step ensures that we measure adversarial robustness rather than errors on already-misranked prompts: we filter out prompts where any evaluated model already scores a rejected response above the chosen response, then sample 100 prompts from the remainder, using the same prompts for all models. This is applied uniformly across all models and does not bias toward any particular model or training method.
>
> **Baselines.** We study PGD, HotFlip, rudimentary perturbations, content injection, and paraphrasing, covering continuous embedding perturbations, gradient-guided discrete swaps, black-box string edits, and semantic attacks. PGD training in embedding space can be viewed as a form of regularization-based defense. Our adversarial training generates adversarial examples on the fly each batch, so the training signal is strong and adaptive. We chose single-step PGD for computational efficiency, as this works well. The combined training strategy already integrates four complementary methods, each targeting different attack surfaces.
>
> **New RLHF results.** We add broader alignment evaluations (OR-Bench, XSTest, WildGuardMix) and a length-controlled analysis; see our response to reviewer ivJn for full tables. Adversarially trained RM policies are consistently preferred on safety benchmarks, and the preference holds even when the base-RM policy produces longer responses, arguing against judge length bias.

---

### Official Review · Reviewer_ivJn · 2026-03-12

**Soundness:** 2
**Presentation:** 2
**Significance:** 2
**Originality:** 2
**Overall Recommendation:** 3
**Confidence:** 3

**Summary:**

This paper studies adversarial robustness in language model systems through a unified view of text scoring models, including dense retrievers, rerankers, and RLHF reward models. The authors argue that robustness research is fragmented across applications and attack types and propose defining adversarial failures as ranking errors, where irrelevant or rejected text receives a higher score than the correct one.
Using this framework, the paper evaluates several adversarial training strategies (e.g., rudimentary perturbations, HotFlip, PGD, content injection, and paraphrasing) and analyzes how robustness transfers across attack types. The results show that most methods mainly defend against the attacks they are trained on, but combining complementary approaches can improve robustness across multiple attack families while maintaining task performance and reducing reward hacking in RLHF.

**Compliance With Llm Reviewing Policy:**

Affirmed.

**Key Questions For Authors:**

The experiments show that adversarial training methods often fail to transfer across attack classes (e.g., token perturbations vs. content injection). Do the authors have insights into the underlying mechanisms behind this lack of transfer?

The combined adversarial training strategy (PGD + HotFlip) is computationally demanding. How does the training cost compare to standard fine-tuning, and how would this overhead scale for larger reward models (e.g., 30B–70B parameters)? Are there approximations that could reduce this cost without sacrificing robustness?

The appendix suggests that robustness does not necessarily scale with model size or effectiveness, with some larger models showing higher success rates for injection attacks. Do the authors have hypotheses explaining why larger models might become more susceptible to such attacks?

The evaluation mainly focuses on specific perturbation and injection attacks. Have the authors considered evaluating robustness against semantic adversarial attacks or multi-turn prompt injection attacks?

In real-world retrieval or ranking systems, attackers often operate in a black-box setting with limited queries. How would the proposed training methods perform under such realistic constraints?

Do adversarially trained reward models maintain their robustness when used with policy models different from those used during RLHF training?

The combined training approach still results in high ASR in several configurations (e.g., HotFlip attacks on retrievers). How do the authors define a practical robustness threshold for deployment, and would they consider the current system secure against a realistic adversary?
Paraphrasing training shows weak and inconsistent robustness gains and was excluded from the combined strategy. Did the authors investigate why paraphrasing fails for e.g., because paraphrases do not resemble true adversarial perturbations or because the score-alignment objective conflicts with ranking objectives?

The RLHF evaluation relies on an LLM judge to measure preference improvements. Since LLM judges can exhibit biases (e.g., toward longer or stylistically different responses), were controls applied for factors such as response length across conditions?

The content-injection threat model assumes adversaries can insert text into passages or responses. For reward models specifically, what is the realistic attack surface (e.g., evaluation-time manipulation, training data poisoning, or exploitation during RLHF rollouts)?

Since the framework unifies text scoring models through score-based adversarial training, have the authors considered extending this approach to directly improve the generative objective of policy models during RLHF?

**Limitations:**

Yes. The authors discuss limitations and dual-use risks in the impact statement, including the possibility that the attack implementations could be misused to manipulate retrieval or ranking systems. They also acknowledge that no adversarial training method achieves perfect robustness and emphasize the need for further research.

**Strengths And Weaknesses:**

#Strengths

The paper introduces a unified framework for studying adversarial robustness across retrievers, rerankers, and RLHF reward models by framing them as text scoring systems and defining failures as ranking errors.

The work includes extensive and rigorous experiments across multiple attack families and adversarial training methods, using strong attack budgets to provide a realistic robustness evaluation.

The analysis provides insightful findings on robustness transfer, showing that defenses often generalize poorly across attack types and revealing important blind spots in current methods.

The paper demonstrates practical impact through RLHF experiments, showing that adversarially trained reward models reduce reward hacking and improve alignment.

#Weaknesses

The combined adversarial training method is computationally demanding because it requires gradient computation for both PGD perturbations and HotFlip-swapped texts. This increases training cost and may limit practicality when scaling to larger reward models.

The proposed training strategy introduces multiple objectives and hyperparameters, but the paper does not thoroughly analyze training stability or sensitivity to these choices. In particular, hyperparameters tuned on the 3B reward model are directly transferred to the 8B model without re-tuning, leaving open questions about robustness to scaling.

Although the experimental evaluation is extensive, the work remains largely empirical. The paper provides limited theoretical or conceptual explanation for why some adversarial training methods transfer across attack types while others fail.

Despite consistent improvements, the absolute robustness gains remain modest in many settings, with high attack success rates persisting for several attacks. As a result, it is not entirely clear how meaningful the observed improvements are for real-world deployment.
The evaluation focuses primarily on token-level perturbations and content injection attacks. Other realistic adversarial strategies, such as semantic adversaries or multi-turn prompt injection attacks, are not explored.

The RLHF experiments are promising but somewhat limited in scope. They rely mainly on LLM-judge evaluations and a small set of benchmarks, and additional human evaluation or broader alignment benchmarks would strengthen the claims about improved downstream alignment.

---

> ### Author Rebuttal · Authors · 2026-03-29
>
> Thank you for your review. We address each point and add new RLHF experiments below.
>
> **RLHF evaluation.** Our RLHF claims rest on KL-divergence analysis and LLM-judge evaluations. We add new broader alignment results (OR-Bench, XSTest, WildGuardMix) and a length-controlled analysis. Judges are Nemotron 3 Super (120B) for the new results and Gemini 3 Flash for Arena-Hard/WildBench. Each judge receives task-appropriate context (harm labels for safety benchmarks, checklists for WildBench). Multiple independent judges across diverse benchmarks, combined with KL-divergence analysis, provide strong evidence.
>
> (OR-Bench+XSTest+WildGuardMix):
>
> |Cmp|KL|W/T/L|
> |--|--:|--:|
> |base>instruct|0.01|65.7/13.5/20.8|
> |base>instruct|0.02|63.4/16.3/20.4|
> |med>base|0.01|44.4/23.8/31.7|
> |med>base|0.02|44.1/26.6/29.4|
> |high>med|0.01|42.1/28.6/29.3|
> |high>med|0.02|39.7/32.8/27.5|
> |high>base|0.01|48.9/25.6/25.5|
> |high>base|0.02|48.4/25.8/25.8|
>
> Adversarially trained RM policies are consistently preferred over base RM policies on these safety benchmarks.
>
> Length-controlled Arena-Hard+WildBench:
>
> |Cmp|KL|Group|W/T/L|
> |--|--:|--|--:|
> |med>base|0.01|all|46.3/11.1/42.6|
> |med>base|0.01|base longer|47.7/10.1/42.2|
> |high>base|0.01|all|46.4/11.0/42.5|
> |high>base|0.01|base longer|48.0/12.5/39.4|
> |med>base|0.02|all|46.1/14.0/39.9|
> |med>base|0.02|base longer|43.1/14.9/42.0|
> |high>base|0.02|all|45.8/15.3/38.9|
> |high>base|0.02|base longer|43.8/19.8/36.4|
>
> Even when the base-RM policy response is longer, adversarially trained RM policies are still preferred, arguing against judge length bias.
>
> **Cost/scaling.** Combined training is our most expensive setting but remains practical: ~20h on 1xH100 (retriever), ~40h on 4xH100 (reranker), ~9h on 4xH100 (3B RM), and ~20h on 4xH100 (8B RM). The extra cost buys broader robustness and better task effectiveness. We have not trained 30B–70B models, but the recipe scales straightforwardly: perturbed texts are additional finetuning data, and PGD/HotFlip add only one gradient step per example.
>
> **Hyperparameter sensitivity.** Hyperparameter transfer is a strength of our approach: without any re-tuning, the 3B hyperparameters transferred to 8B yield the best dev loss and average effectiveness among all adversarial variants. Tables 4–6 extensively sweep training strengths for each method and show stable trends.
>
> **Transfer mechanisms and model size.** The observed transfer patterns follow a clear distinction: Rudimentary/HotFlip/PGD target local token or embedding perturbations, whereas content injection adds text that changes overall meaning. Consistent with this, HotFlip/PGD transfer weakly to injection, while injection-targeted training is the most reliable defense against injection. This motivates combined training. Intuitively, token-level defenses learn local embedding invariances, while injection defenses must learn to discount globally irrelevant text, which is a different inductive bias.
>
> Robustness also does not automatically follow from model size: without explicit robustness training, larger models may still over-score injected but superficially relevant text.
>
> **Practical significance.** We measure robustness not only by ASR but by the number of adversarial edits required for success. Even when ASR remains nonzero, increasing the required attack budget raises the cost and detectability of attacks. This is itself a meaningful robustness gain. Appendix G argues that raw ASR alone is misleading under strong search budgets.
>
> **Black-box and limited-query setting.** Content injection requires no access to the target model. Most of our attacks are black-box (all except HotFlip), and we cover a diverse range of attack budgets across perturbation families. By increasing the edits needed for success, adversarial training raises the cost for any attacker, including query-limited ones, which is practically a success for robustness.
>
> **RM attack surface.** Adversarial texts can realistically arise when adversaries tamper with prompts/context at inference time, during RLHF rollouts when the policy learns to create reward-hacking text, or through poisoned preference data.
>
> **Semantic and multi-turn attacks.** MLM-guided swaps produce naturalistic perturbations via contextual replacements. Our beam search is inherently multi-step, iteratively perturbing over up to 512 steps. Notably, our reward models also process multi-turn conversations, so our attacks already operate on multi-turn inputs.
>
> **Paraphrasing.** We include paraphrasing because ReWordBench advocates it for RM robustness but never evaluates transfer to adversarial attacks or RLHF. Our results confirm it yields weak and inconsistent gains.
>
> **Extending to the generative objective.** Appendix F proposes combining our perturbation families with DPO-inspired constraints. However, we leave this for future work. Our RLHF experiments already show that adversarially robust scoring improves generative outcomes.

---

### Official Review · Reviewer_GyjM · 2026-03-13

**Soundness:** 3
**Presentation:** 3
**Significance:** 3
**Originality:** 3
**Overall Recommendation:** 4
**Confidence:** 3

**Summary:**

The paper study of adversarial robustness in language models, they used text scoring models like retrievers and re-rankers because open-ended generation is not directly testable for adversarial attacks, while scoring models are testable by considering that attacks succeed if an irrelevant text is retrieved or given a higher score than a relevant one. They argue that studying scoring models is better for studying adversarial robustness of language models in general. They evaluate the method on several attack types and evaluate a model for each task of the scoring models, like E5 BERT for the retriever.

**Compliance With Llm Reviewing Policy:**

Affirmed.

**Final Justification:**

The paper is strong in originality and technical soundness. Thus, I have increased my significance score from 2 to 3.

**Key Questions For Authors:**

NA

**Strengths And Weaknesses:**

Strengths:

---

- The paper proposes a method to study adversarial robustness in text scoring models, which is a promising and crucial direction.

- They evaluate the method on a variety of Attack types such as Rudimentary, HotFlip Swaps, and MLM Swaps.

- The claim about open-ended generation methods having an unbounded space of undesirable outputs is correct, and the framing of the idea to use text scoring tasks for LM adversarial robustness is effective and valid since these tasks are quantifiable.

- They evaluate the method in the three scoring models tasks: Retriever, Reranker, Reward.

---

Weaknesses:

---

- One of the claims in the paper is “we introduce multiple adversarial training methods for text scorers,” but the methods used are well-known. What I see is that you did an evaluation of these existing techniques on text scorer models, not introducing new adversarial training methods, even if they have not been used before in text scorer models.

- I think Adversarial robustness in language models is generally dependent on the architectures; however, in your experiments, only one model for each scoring model/task is used. Did you try other models, and the gains still hold? Why did you choose to evaluate only on these models?

- A concern to me is that the paper argues that “studying adversarial robustness in text scoring models is a principled approach for studying the adversarial robustness of language models in general”. All experiments are on text scoring models like retrievers and re-rankers. I understand that the main paper objective is to test and evaluate those methods, but this claim is not validated or proven. It would make the paper stronger if the robustness translates between text scoring tasks and open-ended generation tasks.

- The paper argues that adversarial robustness in language models is fragmented, which is mostly true, and studies the attacks/threats in a unified view. In the Combined Training strategy, my main concern is that it doesn’t always hold or it doesn't always improve the performance, “specialized methods occasionally outperform the combined approach on their sole targeted attacks.” Even when outperforming, the gains are not always significant.

---

> ### Author Rebuttal · Authors · 2026-03-29
>
> Thank you for your positive review. We address each point below.
>
> **Novelty of adversarial training methods.** Our novelty claim is specifically about three points: (1) the adaptation and first systematic evaluation of adversarial training methods for text scoring models, where adapting these methods for robust text scoring is itself non-trivial; (2) adversarial training against content injection, which is entirely new, as no prior work studies training for robustness against content injection in retrievers, rerankers, or reward models; and (3) the unified study yielding novel findings about transfer and combination for these text scoring models, including that robustness transfers poorly across attack types, that combining complementary methods yields broader robustness while even improving effectiveness, and that these adversarially trained reward models reduce reward hacking in RLHF.
>
> **Model diversity.** Our experiments span E5 BERT-base (encoder-only, ~110M params, bidirectional attention) for retrieval, Qwen3-0.6B (decoder-only) for reranking, and Llama-3.2-3B / Llama-3.1-8B (decoder-only, different Llama generations) for reward modeling. This covers encoder vs. decoder architectures, bidirectional vs. causal attention, and 110M to 8B parameters. Crucially, our key findings hold consistently across all of these: robustness transfers poorly across attack types, combined training yields the broadest robustness, and effectiveness is often improved with our training methods. Furthermore, hyperparameters tuned on the 3B reward model transfer directly to 8B with strong results, suggesting the approach generalizes across scale without careful tuning. We also evaluate Skywork-Reward-V2 models at 1B, 3B, and 8B in the appendix, showing that robustness does not automatically improve with scale, reinforcing that adversarial training is needed regardless of model choice or size. We agree that testing on additional models is always desirable, however, we believe this is a meaningfully diverse set given practical compute constraints.
>
> **Principled approach and generalization to generation.** We provide evidence that our attack families generalize to generation: Figure 6 in the Appendix shows that rudimentary manipulations, HotFlip swaps, MLM swaps, and content injection all successfully attack generative LLMs (Gemma-3-27B-it and GPT 5.1). Also consider, adversarially trained reward models directly improve generative LLM quality and alignment. Our RLHF experiments show reduced reward hacking and policies preferred by LLM judges across Arena-Hard, WildBench, and new safety benchmarks (OR-Bench, XSTest, WildGuardMix; see our response to reviewer ivJn for full results). Robust reward models can improve generative LLM robustness during RLHF: they are harder to exploit during training, preventing the policy from learning to produce low-quality or unsafe outputs that game a fragile scorer. Appendix F further proposes a concrete approach for extending our adversarial training framework directly to generative LLMs via DPO-inspired constraints. This is an exciting direction for future work, though beyond the text scoring focus of this paper currently.
>
> **Combined training.** Combined training avoids the main limitation of specialized training: being vulnerable to all other attack types. A single combined model achieves competitive robustness with specialized methods even on their sole targeted attacks, while simultaneously providing robustness across all attack families. In many configurations, the combined model outperforms specialized training even on the attack that specialized training targets. When specialized methods do edge out combined training on their specific attack, the margins are small. Combined training also generally offers better task effectiveness trade-offs, making it often the most practical choice for deployment where attackers can try many strategies. A specialized model that is robust to one attack but remains very vulnerable to all others is not a practical defense.

---

> > ### Author Rebuttal · Reviewer_GyjM · 2026-04-05
> >
> > Thanks for your response regarding the points I have raised. I'll adjust my score accordingly.

---

> > > ### Author Response · Authors · 2026-04-06
> > >
> > > Thank you for your continued engagement. We appreciate your thoughtful consideration of our response and your willingness to revise the score.

---

### Decision · Program_Chairs · 2026-04-30

**Decision:**

Accept (regular)

**Comment:**

The paper's core contribution is to examine how well adversarial training methods work for making language model based scoring methods (such as reward models or rerankers) more robust.

Reviewers liked the framing of the problem as well as the practical impact of the proposed methods to widely-used real-world settings, such as RLHF.

While the reviewers do point out quite a few flaws, overall I believe these flaws are mostly pretty minor. Therefore, I recommend acceptance. The authors should think carefully about the reviewer critiques, however, and improve their writing to make their contributions clearer to a broad ML audience.